# Dnmt3a2/Dnmt3L Overexpression in the Dopaminergic System of Mice Increases Exercise Behavior through Signaling Changes in the Hypothalamus

**DOI:** 10.3390/ijms21176297

**Published:** 2020-08-31

**Authors:** Di Cui, Andrea Mesaros, Gregor Burdeos, Ingo Voigt, Patrick Giavalisco, Yvonne Hinze, Martin Purrio, Bernd Neumaier, Alexander Drzezga, Yayoi Obata, Heike Endepols, Xiangru Xu

**Affiliations:** 1Max Planck Institute for Biology of Ageing, Joseph-Stelzmann-Str. 9b, 50931 Cologne, Germany; Andrea.Mesaros@age.mpg.de (A.M.); gregorburdeos@yahoo.com (G.B.); Ingo.Voigt@age.mpg.de (I.V.); Patrick.Giavalisco@age.mpg.de (P.G.); Yvonne.Hinze@age.mpg.de (Y.H.); Martin.Purrio@age.mpg.de (M.P.); 2Institute for Animal Nutrition and Physiology, Christian Albrechts University Kiel, Hermann-Rodewald Street, 9, 24118 Kiel, Germany; 3University of Cologne, Faculty of Medicine and University Hospital Cologne, Institute of Radiochemistry and Experimental Molecular Imaging, Kerpener Str. 62, 50937 Cologne, Germany; b.neumaier@fz-juelich.de (B.N.); heike.endepols@uk-koeln.de (H.E.); 4Institute for Neuroscience and Medicine, INM-5: Nuclear Chemistry, Forschungszentrum Jülich GmbH, Wilhelm-Johnen-Str., 52425 Jülich, Germany; 5Department of Nuclear Medicine, University of Cologne, Faculty of Medicine and University Hospital Cologne, Kerpener Str. 62, 50937 Köln, Germany; alexander.drzezga@uk-koeln.de; 6Department of Bioscience, Tokyo University of Agriculture, Faculty of Life Sciences, 1-1-1 Sakuragaoka, Setagaya-ku, Tokyo 156-8502, Japan; y1obata@nodai.ac.jp; 7Department of Anesthesiology, Yale University School of Medicine, 10 Amistad Street, New Haven, CT 06519, USA

**Keywords:** DNA methylation, transgenic mouse model, behavioral paradigms, positron emission tomography, dopaminergic system

## Abstract

Dnmt3a2, a *de novo* DNA methyltransferase, is induced by neuronal activity and participates in long-term memory formation with the increased expression of synaptic plasticity genes. We wanted to determine if Dnmt3a2 with its partner Dnmt3L may influence motor behavior via the dopaminergic system. To this end, we generated a mouse line, Dnmt3a2/3L*^Dat^*^/*wt*^, with dopamine transporter (DAT) promotor driven Dnmt3a2/3L overexpression. The mice were studied with behavioral paradigms (e.g., cylinder test, open field, and treadmill), brain slice patch clamp recordings, ex vivo metabolite analysis, and in vivo positron emission tomography (PET) using the dopaminergic tracer 6-[^18^F]FMT. The results showed that spontaneous activity and exercise performance were enhanced in Dnmt3a2/3L*^Dat^*^/*wt*^ mice compared to Dnmt3a2/3L*^wt^*^/*wt*^ controls. Dopaminergic substantia nigra pars compacta neurons of Dnmt3a2/3L*^Dat^*^/*wt*^ animals displayed a higher fire frequency and excitability. However, dopamine concentration was not increased in the striatum, and dopamine metabolite concentration was even significantly decreased. Striatal 6-[^18^F]FMT uptake, reflecting aromatic L-amino acid decarboxylase activity, was the same in Dnmt3a2/3L*^Dat^*^/*wt*^ mice and controls. [^18^F]FDG PET showed that hypothalamic metabolic activity was tightly linked to motor behavior in Dnmt3a2/3L*^Dat^*^/*wt*^ mice. Furthermore, dopamine biosynthesis and motor-related metabolic activity were correlated in the hypothalamus. Our findings suggest that Dnmt3a2/3L, when overexpressed in dopaminergic neurons, modulates motor performance via activation of the nigrostriatal pathway. This does not involve increased dopamine synthesis.

## 1. Introduction

Accumulative studies have demonstrated epigenetic factors as a potent regulator for brain function in the nervous system. Epigenetics encompasses modifying changes in DNA and/or chromatin without altering the basic genetic code, involving DNA methylation, histone post-translational modifications, and post-transcriptional regulation by non-coding RNAs (ncRNAs) [1,2,3]. DNA methylation as one of the major epigenetic processes is mediated by DNA methyltransferases. These enzymes regulate many important cellular processes involved in neuronal activity and brain functions [4,5,6], such as neuron survival, differentiation, memory formation, and several neuropsychological impairments. De novo DNA methyltransferase Dnmt3a2, the shorter isoform of Dnmt3a [7], functions by adding a methyl group onto the C5 position of cytosine to form 5-methylcytosine, the dominant form of DNA methylation in mammals [8]. Dnmt3L, the non-enzymatic cofactor, directly enhances Dnmt enzyme activity [9] and specifically interacts with Dnmt3a2 in the nucleus to stimulate regional DNA methylation in mouse embryonic stem cells [10]. Moreover, Dnmt3L is specifically required by Dnmt3a2 upon functioning in mouse gonocytes [9], indicating the pivotal role of Dnmt3L for the function of Dnmt3a2. Recently, the role of Dnmt3a2 in the brain was revealed. It was reported that Dnmt3a2 expression can be transiently induced by neuronal activity, which in turn induces the expression of synaptic plasticity genes and memory formation [11,12,13]. Enhanced fear-conditioning in old mice was observed during transient over-expression of Dnmt3a2 in the hippocampus. In contrast, transient knock-down of Dnmt3a2 resulted in cognitive deficits [11], indicating that Dnmt3a2 could be a key modulator in regulating neuronal functions regarding learning and cognition. In addition, Dnmt1 and Dnmt3a1 double knock-out mice, as well as single Dnm3a knock-out mice have learning deficits [14,15]. A key regulator in specific synaptic changes is dopamine (DA), particularly in the hippocampal-prefrontal working memory network [16]. We therefore wanted to determine whether Dnmt3a2 with its cofactor Dnmt3L (Dnmt3a2/3L) influences DA activity. To this end, a dopaminergic neuron-specific Dnmt3a2/3L overexpression transgenic mouse line (Dnmt3a2/3L*^Dat^*^/*wt*^) was generated. The dopaminergic neurons of the substantia nigra pars compacta (SNc) have been intensively studied and their electrophysiological properties are well known [17]. Their firing characteristics are homogeneous, therefore even subtle deviations from the common pattern can be detected. For this reason, we focused our study on the nigrostriatal dopaminergic projection. Nigrostriatal DA neuronal activity is associated with motor functions. Disturbances of the DA system are involved in neurological diseases with locomotor impairments, such as Parkinson’s disease, Huntingdon’s disease, and attention deficit and hyperactivity disorder [18]. In Parkinson’s disease, DNA methylation regulates levodopa-induced dyskinesias [19]. We therefore hypothesized that Dnmt3a2/3L may affect the firing patters of DA neurons, which may lead to altered locomotion. We studied the mice with a combination of brain slice electrophysiology, behavioral experiments, in vivo positron emission tomography (PET), indirect calorimetry, and post-mortem metabolite extraction.

## 2. Results

### 2.1. Key Findings

#### 2.1.1. Dnmt3a2/3L Increases Spontaneous Activity and Exercise Performance of Mice with a Gender Preference

In order to generate dopaminergic neuron-specific Dnmt3a2/3L overexpression mouse cohorts (Dnmt3a2/3L*^Dat^*^/*wt*^), the cre/loxp system (Appendix A) was used to cross flox-Dnmt3a2/3L females [20] with Dat-cre males [21], and the offspring were further characterized by DAB immunohistochemical staining (Figure 1A) and immunofluorescense staining of the mCherry reporter (Figure 1B), and the endogenous fluorescence signal can be taken as background (Appendix A). The expression of Dnmt3a2 of Dnmt3a2/3L*^Dat^*^/*wt*^ was increased in the olfactory bulb (OB), striatum (ST) and mid brain (MB) where the DA neurons located, but not in other parts of the brain, e.g., frontal cortex (FC) (Figure 1C). Due to the low endogenous level of Dnmt3L and the expression of Dnmt3L was only detectable by western blot in OB (Appendix A), the transcriptional levels of Dnmt3a2 and Dnmt3L were all significantly increased in midbrain of Dnmt3a2/3L*^Dat^*^/*wt*^ mice (Figure 1D). Although the global methylation at tissue level was not increased (Appendix A), the relative methylation level (normalized to TH) of DA neurons in Dnmt3a2/3L*^Dat^*^/*wt*^ animals was significantly higher than in Dnmt3a2/3L*^wt^*^/*wt*^ mice as anticipated (Figure 1F), and a higher fluorescence intensity in TH positive cells was also detected (Figure 1E). After the characterization, the cylinder test, open field, treadmill and PhenoMaster were performed to test the general activity, locomotion, and metabolism effects on the cohort of Dnmt3a2/3L*^Dat^*^/*wt*^ mice.

The spontaneous rearing activity of the mice was measured by the cylinder test. The number of rearings of Dnmt3a2/3L*^Dat^*^/*wt*^ mice were on average 20% higher compared to controls in mixed gender groups (Figure 1G left). While differences between the female groups reached statistical significance (*p* = 0.0146, Figure 1G middle), the male group showed the same trend (*p* = 0.0947; Figure 1G right). Corresponding to the cylinder test, in the open field assessment female Dnmt3a2/3L*^Dat^*^/*wt*^ mice showed a significant increase of spontaneous activity for total distances travelled compared to the control littermates (*p* = 0.0434; Appendix A). Specifically, the horizontal movement was increased on average by 21.8% for Dnmt3a2/3L*^Dat^*^/*wt*^ mice (*p* = 0.0359; Appendix A). The vertical movement and center stay, however, were not changed (*p* = 0.5096; 0.3711; Appendix A). No significant differences were detected between the male groups (Appendix A). Apart from spontaneous movement, the mice were also forced to run on the treadmill. In mixed gender groups, Dnmt3a2/3L*^Dat^*^/*wt*^ animals ran on average 14.2% further than control littermates (Figure 1H left). If analyzed separately, the difference between the male groups was statistically significant (*p* = 0.0151, Figure 1H right) with Dnmt3a2/3L*^Dat^*^/*wt*^ males running approx. 18.6% further than control males. Although the female groups were not significantly different, they showed the same trend (*p* = 0.1291; Figure 1H middle). Energy metabolism analysis (metabolic cage) showed that Dnmt3a2/3L*^Dat^*^/*wt*^ females had a higher cumulative food (F(1,13) = 10.81; *p* = 0.0059; Appendix A) and water intake (F(1,13) = 4.797, *p* = 0.0474; Appendix A) and a mildly higher respiration quotient especially during daytime (F(1,12) = 4.668; *p* = 0.0571; Appendix A), while male groups remained the same as Dnmt3a2/3L*^wt^*^/*wt*^. The relative energy expenditure in mixed gender groups showed no significant difference (Supplementary Appendix A).

#### 2.1.2. Dopaminergic Neurons from Dnmt3a2/3L*^Dat^*^/*wt*^ Animals Have a Higher Firing Frequency and Excitability

We next determined whether SNc DA neurons of Dnmt3a2/3L*^Dat^*^/*wt*^ mice have altered intrinsic electrophysiological properties. To this end, we assessed several key electrophysiological parameters in the brain slice, such as spontaneous firing rate, precision of pace-making, input resistance, and excitability, which revealed significant differences between DA neurons of Dnmt3a2/3L*^Dat^*^/*wt*^ mice and controls (Figure 2A–F). Overall, SNc DA neurons of Dnmt3a2/3L*^Dat^*^/*wt*^ mice were more active and excitable than those of Dnmt3a2/3L*^wt^*^/*wt*^ littermates. In Dnmt3a2/3L*^Dat^*^/*wt*^ mice, the spontaneous action potential frequency was significantly higher (*p* = 0.048; Figure 2B), and the neurons were more excitable in response to depolarizing current injections, which is reflected in the steeper slope of the current-frequency plots (F(1,24) = 468.7; *p* < 0.0001; Figure 2F). In line with these effects we found an increased cell input resistance (*p* = 0.0006; Figure 2D). Male and female group shown the same tendency (Appendix A). The spontaneous action potential frequency was increased, but also the coefficient of variation, which reflects the lower precision of the pacemaker (*p* = 0.025; Figure 2C). Taken together, the results strongly suggested that spontaneous neuronal activity was upregulated in DA neurons of Dnmt3a2/3L*^Dat^*^/*wt*^ animals.

#### 2.1.3. Dnmt3a2/3L Overexpression Decreased Dopamine Metabolites in the Striatum

The results of metabolite extraction analysis showed that striatal DA levels were the same in Dnmt3a2/3L*^Dat^*^/*wt*^ and Dnmt3a2/3L*^wt^*^/*wt*^ animals (Figure 3). However, tissue concentrations of DA degradation products (3-methoxytyramine [3MT], 3,4-dihydroxyphenyl acetic acid (DOPAC), and homovanillic acid (HVA)) were down-regulated in Dnmt3a2/3L*^Dat^*^/*wt*^ animals, indicating that the degradation of dopamine was slower compared to controls.

#### 2.1.4. Dnmt3a2/3L Overexpression Increased Dopamine Synthesis in Several Brain Areas, But Not in the Striatum

In addition to ex vivo tissue DA and metabolite analysis, we performed in vivo PET using the dopaminergic tracer 6-[^18^F]F-meta-tyrosine (FMT). 6-[^18^F]FMT-PET (image intensity normalized to occipital cortex) delineated the striatal nuclei as expected, but also showed high uptake in the thalamus (Figure 4A,B). There was a tendency of a higher global 6-[^18^F]FMT uptake in Dnmt3a2/3L*^Dat^*^/*wt*^ mice (1.26 ± 0.06 SUV_occ_ vs. 1.16 ± 0.12 SUV_occ_ in the whole brain; *p* = 0.0992; Figure 4D). In the striatum, there was no significant difference in 6-[^18^F]FMT uptake between Dnmt3a2/3L*^Dat^*^/*wt*^ and control mice, neither on the voxel level, nor within a striatal VOI (1.66 ± 0.14 SUV_occ_ vs. 1.56 ± 0.18 SUV_occ_ in the striatum; *p* = 0.2843; Figure 4E). Voxel-wise analysis showed that 6-[^18^F]FMT uptake in Dnmt3a2/3L*^Dat^*^/*wt*^ mice was significantly higher in the left ventral pallidum, left piriform cortex, left sensory cortex (S1 and S2 indicated in Figure 4C) and left amygdala (Figure 4C), compared to controls. Furthermore, a significantly higher symmetric 6-[^18^F]FMT uptake was found in the hypothalamus of Dnmt3a2/3L*^Dat^*^/*wt*^ mice (Figure 4C).

#### 2.1.5. Hypothalamic Metabolic Activity Was Tightly Linked to Motor Behavior in Dnmt3a2/3L*^Dat^*^/*wt*^ Mice, and Correlated to Dopamine Biosynthesis

Awake uptake of ^18^F-labeled glucose ([^18^F]FDG) with a subsequent PET scan was performed in two conditions, home cage resting and easy treadmill running. [^18^F]FDG is trapped in brain cells, therefore our [^18^F]FDG images reflect glucose metabolism between injection and anesthesia, i.e., during home cage stay or running (Figure 5A,B). Some parts of the brain could not be analyzed (shaded areas in Figure 5), because there was considerable spill-over of radioactivity from the head muscles in the home cage condition. This happened because mice groomed, licked, and gnawed, which caused high [^18^F]FDG uptake in the head muscles. On the treadmill, the mice rather focused on running and did not engage their head muscles. The slight spill-over seen in Figure 5B, section +1.0 originated in the Harderian glands. A comparison of the two conditions across all animals revealed that, during running, [^18^F]FDG uptake was significantly higher in the dorsal hippocampus, pretectum and cerebellum (deep cerebellar nuclei and vermis) (red voxels in Figure 5C). A significantly lower [^18^F]FDG uptake during running was found in sensory cortical areas, striatum, and ventral hippocampus (blue voxels in Figure 5C).

Next, we subtracted the home cage image from the treadmill image for every animal and compared the resulting difference images “treadmill minus home cage” beetween groups (Figure 5D). For Dnmt3a2/3L*^Dat^*^/*wt*^ mice, “treadmill minus home cage” was greater in the hypothalamus (dorsomedial and retromammillary nuclei). For control animals, “treadmill minus home cage” was greater in the periaqueductal gray and the right cerebellum (deep cerebellar nuclei). While this result suggests that behavior-related activity changes differed between groups, it does not tell us whether this difference was driven by the home cage condition, the treadmill condition, or both. We therefore analyzed the two conditions separately.

During the home cage condition (Figure 5E) [^18^F]FDG uptake in Dnmt3a2/3L*^Dat^*^/*wt*^ mice was higher in the right somatosensory cortex, the periaqueductal gray, and the right deep cerebellar nuclei, compared to control mice. In control mice, home cage [^18^F]FDG uptake was higher in the right striatum and the hypothalamus (ventromedial nucleus and medial preoptic area). During treadmill running (Figure 5F), [^18^F]FDG uptake in Dnmt3a2/3L mice was higher in the right ventral diagonal band of Broca and the right hypothalamus (lateral preoptic area).

To investigate whether the observed metabolic differences between groups were related to changes in the dopaminergic system, a correlation analysis was performed between the [^18^F]FDG difference images “treadmill versus home cage” and [^18^F]FMT uptake (Figure 5G). We found a positive correlation in the left somatosensory cortex and the hypothalamus (mammillary nuclei). A negative correlation was found in the habenula and pretectum.

## 3. Discussion

The role of Dnmts were so far mostly investigated in mammalian development [22]. Previous studies demonstrated that Dnmts including Dnmt1 and Dnmt3a are required for plasticity in adult forebrain neurons [14], and that hippocampal Dnmt3a2 is essential for cognitive functions in adult mice [11,12,13]. In our study, we uncovered a novel role of Dnmt3a2/3L for regulation of locomotor function and spontaneous activity specifically in the population of dopaminergic neurons.

We showed that Dnmt3a2/3L overexpression in DA SNc neurons increases the firing frequency and excitability of SNc DA neurons from Dnmt3a2/3L*^Dat^*^/*wt*^ mice. SNc neurons project to the striatum, forming the nigrostriatal pathway. A key feature of these neurons is their steady, autonomous pacemaking which maintains extracellular DA levels necessary for basal ganglia network operation [23]. We hypothesized that because of the higher SNc fire frequency, locomotor activity would be increased as well. Indeed, we found higher spontaneous motor activity of young adult Dnmt3a2/3L*^Dat^*^/*wt*^ mice in cylinder and open field tests. This indicates that Dnmt3a2/3L overexpression regulates motor activity on the behavioral level through neuronal activity of nigrostriatal DA neurons.

We further expected that higher SNc fire frequency would increase tonic dopamine release and thus dopamine tissue concentration and/or metabolite concentration in Dnmt3a2/3L*^Dat^*^/*wt*^ mice. Interestingly, ex vivo whole tissue analysis revealed that striatal DA concentration was the same in both groups. Furthermore, concentrations of the DA metabolites 3MT, DOPAC and HVA were significantly lower in Dnmt3a2/3L*^Dat^*^/*wt*^ mice, indicating that long-term DA synthesis was not increased. Earlier studies, where a short-term increase of fire frequency was induced by electrical stimulation of the medial forebrain bundle, demonstrated a frequency-dependent release of dopamine [24], increased activity of tyrosine hydroxylase [23] and elevated tissue concentration of DOPAC [25]. However, DA tissue concentration remained constant [25]. A recent study employing deep brain stimulation of the medial forebrain bundle in rats confirmed that striatal DA release was elevated at onset of stimulation [26], but data on long-term effects are still lacking. DA release is strongly controlled by several mechanisms, including dopamine transporter activity (reuptake), D2 autoreceptor activity (inhibition of DA release), glutamatergic activity of neighboring synapses (H_2_O_2_ as transsynaptic messenger), and cholinergic activity (via muscarinergic receptors) [27]. Thus, DA fire frequency provides temporal information, while the quantity of DA release and therefore DA synthesis may be regulated independent from firing rate.

To examine DA synthesis capacity in vivo, we performed PET imaging with the tracer 6-[^18^F]FMT. We have chosen this dopamine analog because it is not a substrate of the degrading enzyme catechol *O*-methyl transferase (COMT), and is only metabolized by amino acid decarboxylase (AADC; the enzyme that converts L-DOPA into DA) and subsequently monoamine oxidase (MAO) [28]. 30 min after injection, most of brain radioactivity is derived from the metabolite 6-[^18^F]fluoro-3-hydroxyphenylacetic acid (6-[^18^F]FPAC) [29]. Because 6-[^18^F]FPAC does not cross the blood–brain barrier, it is trapped in the brain [29]. When peripheral AADC is blocked, e.g., by benserazide, it is assumed that 6-[^18^F]FMT uptake mainly shows cerebral AADC activity with high image contrast [30]. Although AADC is not the rate-limiting step of DA synthesis, it has been shown that the enzyme is regulatable [31]. In the striatum, where dopaminergic input is dominating, AADC activity is thought to reflect presynaptic DA biosynthesis. In extrastriatal areas, such as the thalamus and midbrain, a considerable amount of 6-[^18^F]FMT is taken up by other monoaminergic presynaptic terminals containing noradrenaline and serotonin as transmitters [32]. In addition, monoenzymatic AADC neurons, e.g., in the nucleus of the solitary tract [33], take up 6-[^18^F]FMT as well. Although global 6-[^18^F]FMT uptake is a mixture of DA and non-DA AADC activity, we found a tendency of increased global uptake in Dnmt3a2/3L*^Dat^*^/*wt*^ mice. In the striatum, however, there was no significant difference between groups, indicating that DA biosynthesis was not increased in nigrostriatal presynaptic terminals of Dnmt3a2/3L*^Dat^*^/*wt*^ mice. In contrast, we found a significantly elevated 6-[^18^F]FMT uptake in the left ventral pallidum of Dnmt3a2/3L*^Dat^*^/*wt*^ mice. The ventral pallidum receives DA afferents from both SNc and VTA and is involved in reward signaling [34]. Increased dopamine biosynthesis indicates stronger activation of reward pathways in Dnmt3a2/3L*^Dat^*^/*wt*^ mice.

In addition, 6-[^18^F]FMT uptake was also increased in the hypothalamus and the pituitary gland indicating increased DA signaling in the tuberoinfundibular pathway of Dnmt3a2/3L*^Dat^*^/*wt*^ mice. This pathway originates in the hypothalamic arcuate nucleus and projects to the pituitary gland, where DA inhibits prolactin release [35]. Prolactin is involved in the regulation of energy and metabolic homeostasis [36], and increases proportionally to exercise intensity [37].

The other areas where 6-[^18^F]FMT uptake was significantly increased in Dnmt3a2/3L*^Dat^*^/*wt*^ mice (piriform and sensory cortex) receive only weak DA inputs. We suspect that 6-[^18^F]FMT uptake changes in these brain regions reached statistical significance more easily because of low baseline AADC activity, in contrast to regions where AADC activity is already very high in wildtype mice (e.g., striatum). Taken together, in vivo 6-[^18^F]FMT PET results support our ex vivo tissue analysis finding that striatal DA biosynthesis was not increased in Dnmt3a2/3L*^Dat^*^/*wt*^ mice. We can only speculate about how increased nigrostriatal firing may have led to higher locomotor activity without increasing DA synthesis. When inter-spike intervals are shorter than 200 ms, DA cannot be completely cleared from the synaptic cleft before the the next release [38]. This may lead to an overstimulation of postsynaptic receptors, facilitating movement [39]. Further experiments are needed to investigate the relationship between SNc neuronal activity and locomotion.

Next, we wanted to investigate if neuronal activity in motor regions including the basal ganglia and/or the hypothalamus were altered in Dnmt3a2/3L*^Dat^*^/*wt*^ mice. To this end we used metabolic PET imaging with the tracer [^18^F]FDG, which reflects synaptic activity [40,41,42]. We compared [^18^F]FDG uptake during two behavioral settings, continuous treadmill running and staying in the home cage. This was done with all animals pooled to describe the basic motor patterns. During running, we found increased activity in the medial part of the cerebellum in both groups. This region corresponds to the cerebellar motor region, which is involved in the control of rhythmic movements [43]. The dorsal hippocampus was also activated during running, probably caused by an increase of theta frequency during locomotor behavior [44]. Striatum and sensory cortices were more active during home cage activity, which may reflect random non-automatic motor behaviors such as sporadic walking, grooming, and gnawing. The motor cortex-basal ganglia loops are important for initiation of movement [45], but are no longer needed to maintain rhythmic automatic activity such as running, which leads to a lower activity in the treadmill condition.

For group comparison, we analyzed the difference images “treadmill minus homecage”. They correspond to the pattern described above, i.e., they reflect the individual activity changes between the two behavioral conditions. No significant group differences were present in the areas described above. While this indicates that general motor patterns were the same in Dnmt3a2/3L*^Dat^*^/*wt*^ mice and controls, we found significant differences between groups in other areas. A negative cluster emerged in the right deep cerebellar nuclei and the periaqueductal gray, indicating that “treadmill minus home cage” was greater in controls compared to Dat-Dnmt3a2/3L mice. This was driven by a group difference in the home cage condition, where the deep cerebellar nuclei and the periaqueductal gray were more active in Dnmt3a2/3L*^Dat^*^/*wt*^ mice during their home cage stay compared to controls. However, no group difference was seen in these brain areas during running. This result indicates that brain activation patterns were different between groups when the animals engaged in random home cage activity. These differences disappeared when the mice were involved in treadmill running. The home cage differences may therefore merely reflect the individual behaviors of the mice during the 40 min uptake period, rather than inherent activity differences related to the transgene.

However, this explanation does not account for the hypothalamus. Dnmt3a2/3L*^Dat^*^/*wt*^ mice showed a lower hypothalamic activity in the home cage condition, statistically significant in the medial preoptic nucleus and the ventromedial hypothalamus. During treadmill running, hypothalamic activity was higher compared to controls, significant in the lateral preoptic nucleus. These results suggest that hypothalamic activity was more tightly linked to motor behaviour in Dnmt3a2/3L*^Dat^*^/*wt*^ mice compared to controls. With 6-[^18^F]FMT we have found a higher dopaminergic activity in the hypothalamus, which was correlated to the metabolic activity changes observed with [^18^F]FDG. Hypothalamic dopamine plays a crucial role in central fatigue mechanisms. The serotonin-to-dopamine ratio in the hypothalamus is positively correlated with the time to fatigue [46]. Higher dopamine concentrations, particularly in the preoptic area, block the signal for exercise cessation [47] and lead to higher motor performance [48]. The higher motor activity of Dat-Dnmt3a2/3L mice may therefore be caused by increased dopamine signaling in the hypothalamus rather than in the nigrostriatal system.

## 4. Materials and Methods

### 4.1. Transgenic Mice

Dnmt3a2/3L*^Dat^*^/*wt*^ mice were generated by mating male Dat cre mice with female Dnmt3a2/3L*^flox^*^/*flox*^ mice. They were 3–5 months old and had a body weight of 18–32 g. The mice were housed in groups with ad libitum food and water supply in a 12h light/dark cycle. There were two groups, Dnmt3a2/3L*^Dat^*^/*wt*^ mice (*n* = 60) with Dnmt3a2/3L overexpression in the DA system, and Dnmt3a2/3L*^Dat^*^/*wt*^ littermate controls (*n* = 60). Animals were assigned to their group after genotyping the ear notches accrueing during ear tagging. Experiments were carried out in accordance with the EU directive 2010/63/EU for animal experiments and the German Animal Welfare Act (TierSchG, 2006), and were approved by regional authorities (LANUV NRW, approval number 84-02.04.2016.A317, approved on 9 March 2017).

### 4.2. Immunohistochemistry

Brains from Dnmt3a2/3L*^Dat^*^/*wt*^ mice and Dnmt3a2/3L*^wt^*^/*wt*^ littermate controls were harvested, and embedded in OCT. Brain sections (30 μm) were immunostained with primary antibody overnight at 4 °C, washed PBS (6 × 5 min), then incubated with secondary antibody for 2 h at room temperature, washed with PBS (6 × 5 min), and mounted using mounting gel (Thermo Fisher, P36965, Waltham, MA, USA). The next day, the slices were visualized under a fluorescence microscope. For DAB staining, after washing, the brain sections were incubated with the secondary antibody, and the HRP reagent for 1h. Reactions were visualized by developing DAB substrates and were visualized under a light microscope.

### 4.3. DNA Methylation Level

The immunofluorescence intensity on single cell based of 5mC and TH was measured by Image J from three Dnmt3a2/3L*^Dat^*^/*wt*^ and Dnmt3a2/3L*^wt^*^/*wt*^ mice. The DNA methylation level of each TH positive cells was normalized by ratio of 5mC/TH. The unpaired *t*-test was used to compare Dnmt3a2/3L*^Dat^*^/*wt*^ mice and Dnmt3a2/3L*^wt^*^/*wt*^ mice.

### 4.4. Behavior and Metabolism

#### 4.4.1. Cylinder Test

The spontaneous activity of the mice can be measured by the cylinder test. The mouse was placed in a cylinder (15.5 cm diameter), and the activity of the mouse was recorded for 3 min using a video camera. The number of rearings was evaluated. The protocol was similar as in [49].

#### 4.4.2. Open Field

Activity was measured in a 10 min session in a transparent open field box of 50 cm × 50 cm × 40 cm (Locomotor Activity and Open Field, TSE Systems GmbH, Bad Homburg, Germany). The distance travelled, horizontal and vertical movement, and the time in the central zone (18.5 cm × 18.5 cm) were recorded every minute. The unpaired *t*-test was used to compare Dnmt3a2/3L*^wt^*^/*wt*^ and Dnmt3a2/3L*^Dat^*^/*wt*^ mice.

#### 4.4.3. Treadmill

Mice were placed on the treadmill (TSE systems GmbH, Bad Homburg vor der Höhe, Germany) for a five-minute habitation. After a ten-minute warm-up at the speed of 0.1 m/s, the speed increased continuously by 0.02 m/s per minute. When a mouse stopped moving, the belt carried it backwards on a shock grid. We used a low current of 0.3 mA according to the standard protocol [39]. The distance moved was recorded until three consecutive electroshocks were received.

#### 4.4.4. Metabolic Cages

The energy homeostasis of mice was measured by indirect calorimetry, using metabolic cages (Phenomaster, TSE Systems GmbH) for 4 days. Before the measurement, mice were habituated in training cages for 3–4 days. Subsequently, a further habituation phase of 24 h took place in the actual metabolic cages. Subsequently, food and water intake, calorie consumption, and respiratory quotient were measured for 48 h.

#### 4.4.5. Electrophysiology

The recordings were carried out essentially as described previously [50,51]. Experiments were performed on coronal brain slices from 14–16 weeks old male and female Dnmt3a2/3L^*Dat*/*wt*^ mice (*n* = 3) and Dnmt3a2/3L*^wt^*^/*wt*^ littermate (controls) (*n* = 3). Perforated patch recordings were performed with a modified ELC03-XS amplifier (NPI Electronic, Tamm, Germany) controlled by the software PatchMaster (version 2.32; HEKA, Lambrecht, Germany) via a LIH 1600 data acquisition system (HEKA). The cell input resistance was calculated from voltage responses to small hyperpolarizing current pulses. To analyze excitability, i.e., evoked action potential firing, a series of depolarizing current pulses (15 pA to 210 pA in 15 pA increments; 1.5 s duration) were injected from a holding potential of −70mV. For each current pulse, the number of action potentials was determined. For details, see Appendix A.

#### 4.4.6. Tissue Metabolite Analysis

Mouse striatum was prepared from Dnmt3a2/3L*^Dat^*^/*wt*^ and Dnmt3a2/3L*^wt^*^/*wt*^ mice. The metabolite extraction was performed as described by [52] slight modifications (see Appendix A). Quantitative analysis of dopamine (DA), 3-methoxytyramine (3MT), 3,4-dihydroxyphenyl acetic acid (DOPAC) and homovanillic acid (HVA) was performed using an Acquitiy UPLCTM I-class chromatographic system coupled to a XevoTM TQ-S mass spectrometer (Waters, Elstree, UK).

#### 4.4.7. Positron Emission Tomography (PET)

A Focus 220 micro PET scanner (CTI-Siemens, Munich, Germany) with a resolution of 1.4 mm was used. PET measurements of the DA system were performed for 40 min with intravenously injected 6-[^18^F]FMT in *n* = 6 Dnmt3a2/3L*^Dat^*^/*wt*^ mice (4 m, 2 f) and *n* = 6 Dnmt3a2/3L*^wt^*^/*wt*^ control littermates (4 m, 3 f) in the anesthetized state. For [^18^F]FDG PET, performed with the same animals, the tracer was injected intraperitoneally during a short (1 min) isoflurane anesthesia. The mouse was then transferred to the treadmill which was operated at low speed (0.1 m/s). After 30 min of easy running, the mouse was anestetized with isoflurane and placed in the PET scanner (see above). The emission scan started 40 min after [^18^F]FDG injection and stopped after 30 min of data collection. Images were reconstructed using an iterative OSEM3D/MAP procedure [53]. Voxel sizes were 0.38 × 0.38 × 0.82 mm^3^. All further analysis was done with the Software VINCI 4.92 (MPI for Metabolism Research, Cologne, Germany). For details, see Appendix A.

#### 4.4.8. Quantification and Statistical Analysis

##### Behavior and Metabolism

Cylinder test, Open field and treadmill experiments were analyzed with unpaired Student’s *t*-test, using Graphpad Prism, *p*-values less than 0.05 were considered as statistically significant.

Metabolism data from metabolic cage (respiratory quotient, calorie consumption per lean weight, and food and water intake) were analyzed using two-way ANOVA followed by a Bonferroni multiple comparison test. Factors were genotype (Dnmt3a2/3L*^Dat^*^/*wt*^ and Dnmt3a2/3L*^wt^*^/*wt*^) and time.

##### Electrophysiology

Data analysis of electrophysiological cell recording was performed with Spike2 (Cambridge Electronic Design Limited, Milton, Cambridge, UK), Graphpad Prism (version 5.0b; Graphpad Software Inc., San Diego, CA, USA) and Igor Pro 6 (WaveMetrics, Inc., G-7, Portland, OR, USA). Two-tailed unpaired *t*-tests or Welch’s unpaired *t*-tests were performed to determine differences in electrophysiological properties between SNc DA neurons of Dnmt3a2/3L*^Dat^*^/*wt*^ and Dnmt3a2/3L*^wt^*^/*wt*^ mice. To determine differences in linear regression slopes Graphpad Prism’s linear regression analysis (equivalent to ANCOVA) was used. A significance level of 0.05 was accepted for all tests. 

#### 4.4.9. Tissue Metabolite Analysis

Metabolite concentrations in the two groups were compared with an unpaired Student’s *t*-test, using Graphpad Prism (version 5.0b; Graphpad Software Inc., San Diego, CA, USA).

#### 4.4.10. Positron Emission Tomography (PET)

Voxel-wise comparison between groups was performed with an unpaired Student’s *t*-test using the Software VINCI 4.92 (MPI for Metabolism Research, Cologne, Germany). The resulting statistical map was corrected for multiple testing on the *p* < 0.05 level with a threshold-free cluster enhancement (TFCE) procedure [54].

## 5. Conclusions

We conclude that DNA methylation mediated by Dnmt3a2/3L is able to increase neuronal activity in DA neurons, leading to elevated locomotor activity. This is a first step in understanding the neuronal effects of epigenetic modification and its behavioral consequences. We are aware that the present study has not explicitly elucidated the role of DNA methylation by Dnmt3a2/3L in this significant biological consequence yet. For example, at the tissue level, we hardly obtained any significantly synaptic plasticity gene expression change (Appendix A), although we achieved that in the in vitro study, which is likely due to the low percentage of dopaminergic neurons among all cell populations in the brain tissues. To further decipher the role of DNA methylation as well as the gene expression regulated by Dnmt3a2/Dnmt3L in neuronal functions and behavioral changes, a catalytic domain mutant (Cys-Ala) control mouse cohort is needed to understand the enzyme activity of Dnmt3a2 [55], and the single cell (population) DNA methylation, gene expression, and recording are essential to decode the involving epigenetic and molecular mechanism(s).

## Figures and Tables

**Figure 1 ijms-21-06297-f001:**
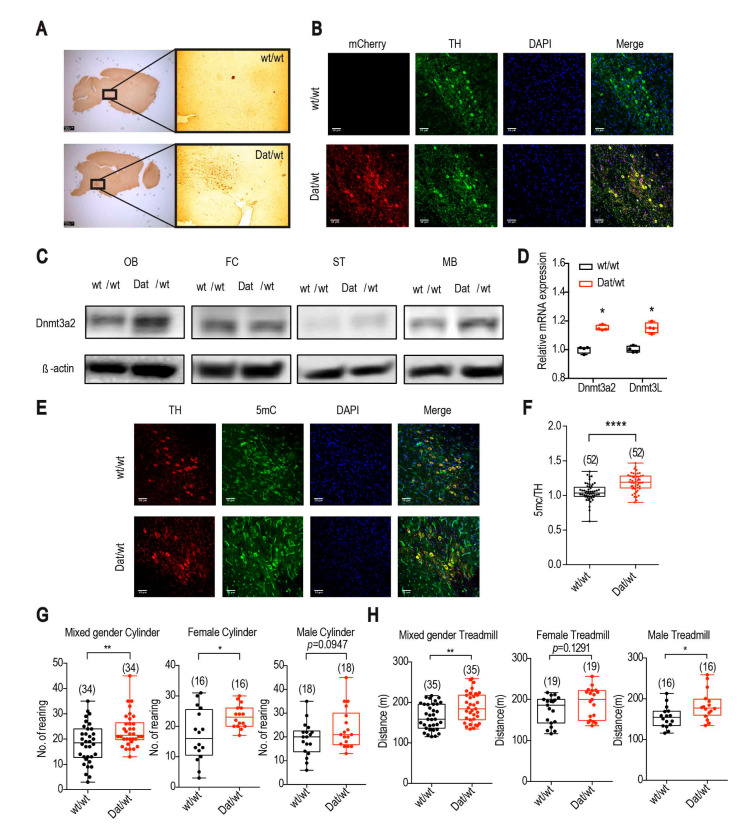
Dnmt3a2/3L specifically overexpressed in dopaminergic neurons, and Dnmt3a2/3L*^Dat^*^/*wt*^ animals showed greater spontaneous activity and higher exercise performance. (**A**) DAB immunohistochemical staining showed that the mCherry reporter was specifically expressed in midbrain areas in Dnmt3a2/3L*^Dat^*^/*wt*^ mice compared to control littermates. (**B**) mCherry was expressed specifically in neurons double labelled for tyrosine hydroxylase (TH) in Dnmt3a2/3L*^Dat^*^/*wt*^ mice. (**C**) Dnmt3a2 expression level detected by western blot was increased in the olfactory bulb (OB), striatum (ST) and midbrain (MB) in Dnmt3a2/3L*^Dat^*^/*wt*^ mice. (**D**) qRT-PCR of Dnmt3a2 and Dnmt3L was shown an elevated expression in midbrain of Dnmt3a2/3L*^Dat^*^/*wt*^ mice. (**E**) 5mC staining showed stronger fluorescence intensity in TH positive cells in Dnmt3a2/3L*^Dat^*^/*wt*^ mice. (**F**) Quantification for the DNA methylation of TH positive cells was significantly higher in Dnmt3a2/3L*^Dat^*^/*wt*^ mice. (**G**) Dnmt3a2/3L*^Dat^*^/*wt*^ mice exhibited a higher spontaneous activity in the cylinder test in mixed gender and female groups, no significance but the same trend was found in the male group. (**H**) Dnmt3a2/3L*^Dat^*^/*wt*^ mice showed a higher exercise performance on the treadmill in mixed gender and male groups, no significance but the same trend occurred in the female group. Values are reported as the mean ± SEM. * *p* < 0.05, ** *p* < 0.01, **** *p* < 0.0001. Student *t*-test has been used for comparison between Dnmt3a2/3L*^wt^*^/*wt*^ and Dnmt3a2/3L*^Dat^*^/*wt*^.

**Figure 2 ijms-21-06297-f002:**
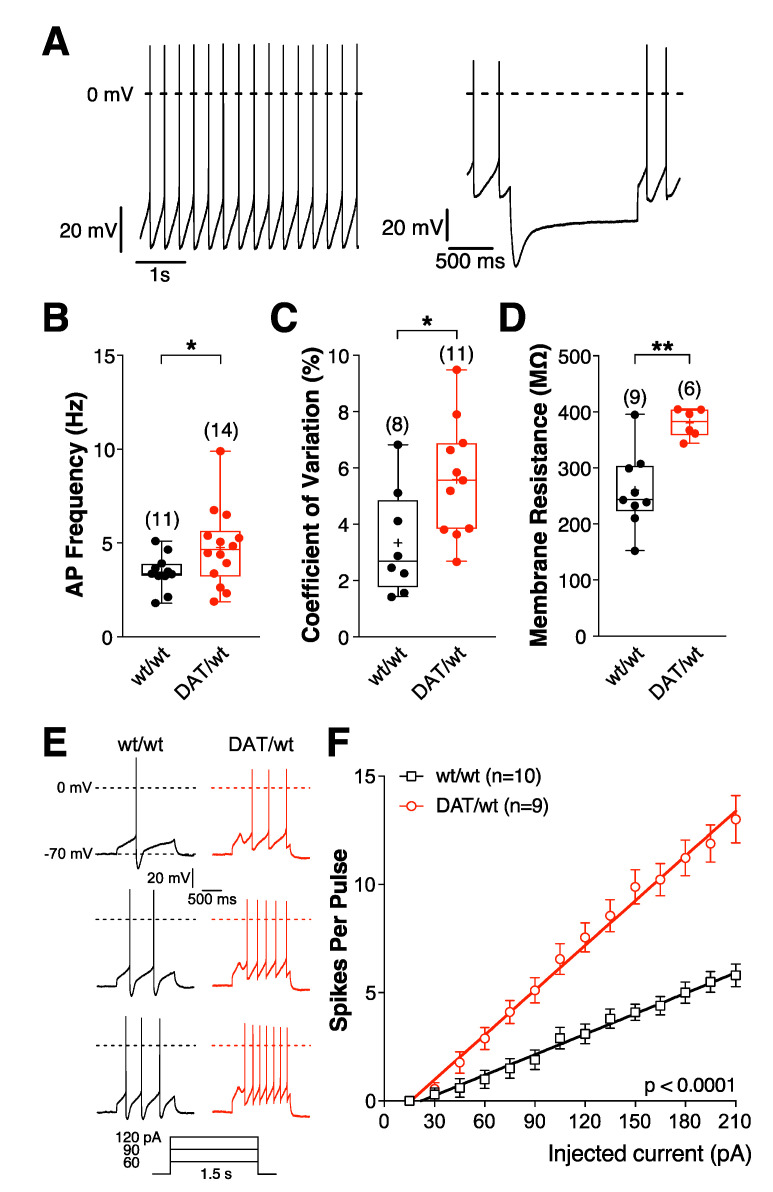
Intrinsic electrophysiological properties of SNc DA neurons are changed in Dnmt3a2/3L*^Dat^*^/*wt*^ mice. (**A**–**F**) Electrophysiological parameters of SNc DA neurons in Dnmt3a2/3L*^Dat^*^/*wt*^ and Dnmt3a2/3L*^wt^*^/*wt*^ mice. Current clamp recordings were performed in the perforated patch clamp configuration. (**A**) DA neurons in the SNc were identified by their slow and regular firing (left) and the presence of a large *I*_H_ (hyperpolarization activated cation current)-dependent ‘sag’ potential (right). (**B**) Spontaneous activity. (**C**) Coefficiant of variation. (**D**) Cell input resistance. * *p* < 0.05, ** *p* < 0.01. *p*-values are for Welch’s unpaired Student’s *t*-tests (**B**,**D**) or unpaired *t*-tests (**C**). Excitability of DA neurons in Dnmt3a2/3L*^Dat^*^/*wt*^ and Dnmt3a2/3L*^wt^*^/*wt*^ mice. (**E**) Representative voltage responses to depolarizing current pulses. (**F**) Mean spike counts during current pulses (1.5 s) as a function of injected current from a holding potential of −70 mV. Mean values are represented as mean ± SEM. F(1,24) = 468.7; *p* < 0.0001.

**Figure 3 ijms-21-06297-f003:**
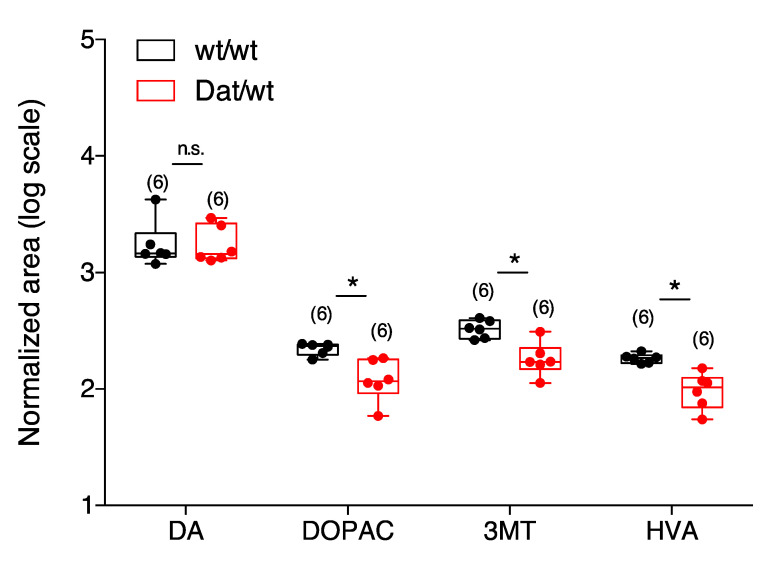
Levels of neurotransmitters in the striatum. The level of dopamine was the same in both Dnmt3a2/3L*^Dat^*^/*wt*^ and Dnmt3a2/3L*^wt^*^/*wt*^ animals. The levels of DOPAC, 3MT, HVA were lower in Dnmt3a2/3L*^Dat^*^/*wt*^ animals. Abbreviations: DA: dopamine; 3MT: 3-methoxytyramine (extracellular metabolite); DOPAC: 3,4-dihydroxyphenyl acetic acid; HVA: homovanillic acid. Values are reported as the mean ± SEM. * *p* < 0.05. Student *t*-test has been used for comparison

**Figure 4 ijms-21-06297-f004:**
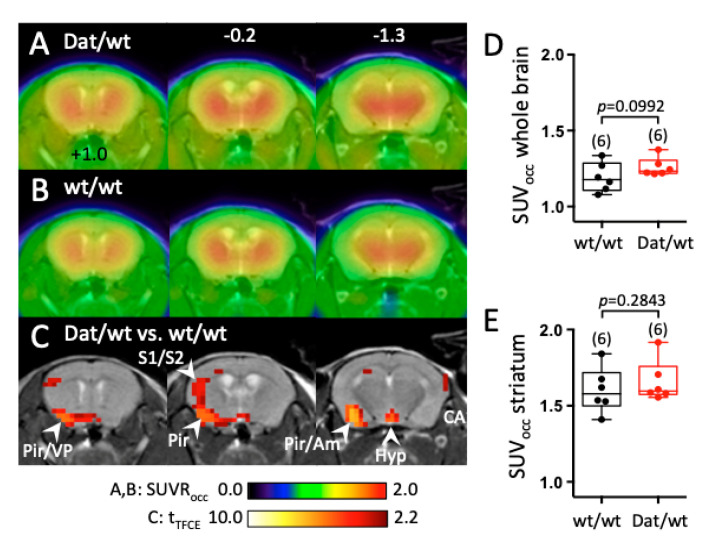
Change in 6-[^18^F]FMT uptake in the brain of Dnmt3a2/3L*^Dat^*^/*wt*^ mice compared to control littermates. (**A**) Mean cerebral 6-[^18^F]FMT uptake in Dnmt3a2/3L*^Dat^*^/*wt*^ mice (*n* = 6). Image intensity was normalized to 6-[^18^F]FMT uptake in the occipital cortex. (**B**) Mean 6-[^18^F]FMT uptake of control littermates (*n* = 6). Both genders were used, and uptake took place under anaesthesia. (**C**) Student’s *t*-test (*p* < 0.05, controlled for multiple testing with a TFCE procedure) for comparison between control littermates and Dnmt3a2/3L*^Dat^*^/*wt*^ mice. Red voxels indicate significantly higher 6-[^18^F]FMT uptake in Dnmt3a2/3L*^Dat^*^/*wt*^ mice. Significantly higher uptake in control mice was not observed. (**D**) Comparison of global 6-[^18^F]FMT uptake between groups. (**E**) Comparison of striatal 6-[^18^F]FMT uptake between groups. Abbreviations: Am: amygdala, Hyp: hypothalamus, Pir: piriform cortex, S1/S2: primary and secondary sensory cortex, VP: ventral pallidum.

**Figure 5 ijms-21-06297-f005:**
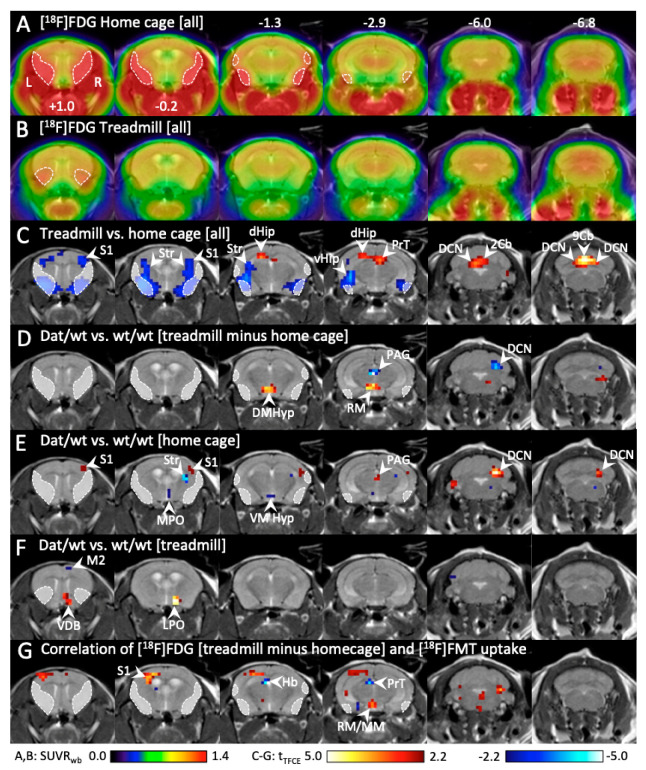
Change in cerebral [^18^F]FDG uptake during running and home cage stay in Dnmt3a2/3L*^Dat^*^/*wt*^ mice compared to control littermates.(**A**) Mean [^18^F]FDG uptake in the homecage (*n* = 14). Because the mice groomed and gnawed, head muscles were active, causing radioactivity spillover into the brain (sketched areas). (**B**) Mean [^18^F]FDG uptake during running on the treadmill (*n* = 14). Head muscles were relaxed, only some spillover (sketched areas) from the Harderian glands is visible. (**C**) Significant differences (*p* < 0.05, corrected for multiple testing) between [^18^F]FDG uptake during home cage and treadmill running across all animals. Red voxels: significantly higher [^18^F]FDG uptake during running. Blue voxels: significantly higher [^18^F]FDG uptake during home cages stay. (**D**) Comparison (*p* < 0.05, corrected for multiple testing) of [^18^F]FDG difference images (“treadmill minus home cage”) between Dnmt3a2/3L*^Dat^*^/*wt*^ mice and controls (*n* = 7 each). Red voxels: significantly higher difference between running and home cage in Dnmt3a2/3L*^Dat^*^/*wt*^ mice. Blue voxels: significantly higher difference between running and home cage in controls. (**E**) Significant differences (*p* < 0.05, corrected for multiple testing) of [^18^F]FDG uptake in the home cage between Dnmt3a2/3L*^Dat^*^/*wt*^ mice and controls (*n* = 7 each). Red voxels: significantly higher [^18^F]FDG uptake in Dnmt3a2/3L*^Dat^*^/*wt*^ mice. Blue voxels: significantly higher [^18^F]FDG uptake in control mice.(**F**) Significant differences of [^18^F]FDG uptake during treadmill running between Dnmt3a2/3L*^Dat^*^/*wt*^ mice and controls (*n* = 7 each).(**G**) Correlation (*p* < 0.05, corrected for multiple testing with TFCE) between [^18^F]FDG difference images (“treadmill minus home cage”) and [^18^F]FMT images with all animals pooled (*n* = 12). Red voxels: positive correlation between [^18^F]FMT uptake and [^18^F]FDG difference images. Blue voxels: negative correlation.Abbreviations: 2Cb, 9Cb: lobule 2 and 9 of the cerebellar vermis, AHyp: anterior hypothalamus, DCN: deep cerebellar nuclei, dHip: dorsal hippocampus, DMHyp: dorsomedial hypothalamus, Hb: habenula, LPO: lateral preoptic nucleus; M2: secondary motor cortex; MM: medial mammillary nucleus, MPO: medial preoptic nucleus; PAG: periaqueductal gray, PFl: paraflocculus, PrT: pretectum, RM: retromammillary nucleus, S1: primary sensory cortex, Sim: simple lobule of the cerebellum, Str: striatum, VDB: ventral diagonal band of Broca, vHip: ventral hippocampus, VMHyp: ventromedial hypothalamus; VTh: ventral thalamus.

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
