# Peer review of "Dnmt3a2/Dnmt3L Overexpression in the Dopaminergic System of Mice Increases Exercise Behavior through Signaling Changes in the Hypothalamus"

_ijms, 2020, doi:10.3390/ijms21176297_

Round 1

Reviewer 1 Report

In this paper, the authors investigate if the de novo methyltransferase DNMT3a2 has any effects on dopaminergic neurons of the SNc. The paper is well written, and the authors’ conclusion – that Dnmt3a2 overexpression increases neuronal activity of the SNc-DA neurons, and that this in turn affects locomotor functions is largely supported by the data presented. However, the scope/interest of the manuscript to a broad audience is somewhat limited, partly due to very little mechanism (something the authors freely admit- to their credit). Given that this is a largely descriptive study, I have certain suggestions that may improve the paper.

Specific Comments:

  1. The authors conclude that the effect they observe is due to increased DNA methylation in the SNc neurons. However, there is not much data in the paper that supports it (except for Fig 1B). A quantitative comparison between the levels of DNMT3a2 and DNA methylation in these mice would be extremely useful to further strengthen the argument that it is indeed the increased DNA methylation that drives the observed effects.
  2. The authors use the mCherry to determine levels of the transgenic DNMT3a2. However, given that the overexpression causes increased neuronal activity, and that increased neuronal activity has previously been shown to drive DNMT3a2 expression/activity, the authors must include a western blot depicting the expression levels of endogenous DNMT3a2.
  3. It would be interesting if the authors can provide some hint of mechanisms – for example, are there any gene expression changes? I realize that a detailed mechanism may be beyond the scope of what the authors would want to include, but a simple qPCR experiment of some target genes may be interesting to add.
  4. Given that there is a striking gender influence for phenotypes in Fig 1, did the authors observe any gender-influences for the neuronal activity experiments in fig 2?
  5. Given the PET results (lines 239-244), do the authors think the phenotypes are due to DA neurons or were the activities of any non-DA neurons affected? Can they test this?
  6. The methods (in the supplement) were incomplete. The authors should elaborate and complete those sections.

Author Response

Dear Reviewer 1,

Many thanks for your comments and suggestions. Please see the attachment as reply.

Best regards,

Authors

Reviewer 2 Report

In this study, Dr Cui et al have generated a mouse line, Dnmt3a2/3LDat/wt, with dopamine 27 transporter (DAT) promotor driving Dnmt3a2/3L overexpression. Both male, female Dnmt3a2/3LDat/wt mice were studied and compared with WT phenotype for 28 behavioral paradigms (e.g., cylinder test, open field and treadmill), brain slice patch clamp 29 recordings, ex vivo metabolite analysis, and in vivo positron emission tomography (PET) using the 30 dopaminergic tracer 6-[18F]FMT. The authors observed that spontaneous activity and exercise performance were enhanced in Dnmt3a2/3LDat/wt mice compared to WT controls. This and metabolite observations led the authors concluded that DNA methylation mediated by Dnmt3a2/3L is able to increase neuronal activity in DA neurons, leading to elevated locomotor activity.

If indeed these observations are valid, this will be the first validation of the neuronal effects of epigenetic modification and its behavioral consequences in mammals. The manuscript is written well, and experiments are performed with correct numbers of mice.

However, I have a real concern in the experimental planning since it lack a catalytic mutant control. Ideally, Cys to Ala mutation at the catalytic site of Dnmt3a2 would have rendered the enzyme inactive. A mouse line, Dnmt3a2 (Cys-Ala)/3LDat/wt, with dopamine 27 transporter (DAT) promotor mediated overexpression should have been compared directly with Dnmt3a2 functional line (Dnmt3a2/3LDat/wt). Any advantage gained in Dnmt3a2/3LDat/wt line would have been lost in mutant line and provided the evidence of DNA hypermethylation participating in behavioral consequences in mammals. Each comparison should have 3 sets of animals wt, Dnmt3a2/3LDat/wt and Dnmt3a2 (Cys-Ala)/3LDat/wt. As the manuscript stands now, it will be impossible to know if the observed behaviors are due to simple overexpression artifacts.

Author Response

Dear Reviewer 2,

Many thanks for your comments and suggestions. Please see the attachment as reply.

Best regards,

Authors

Reviewer 3 Report

The manuscript by Di Cui et al. deals with the characterization of DNA methylation via phenotyping of Dnmt3a2/3L overexpression mouse model in dopaminergic neural system. The centerpiece of this study is well designed and introducing a distinct aspect of epigenetic regulation in neural system. While the questions remain how the DNA methylation disrupt DA neuron activity and their molecular targets, the observation of increasing spontaneous activity and altered dopamine metabolisms are very exciting. There were some concerns with the paper and the followings will help the author to improve their revision:

  1. The mouse strain author used here is flox-Dnmt3a2/3L in Ref.17, which contains GFP reporter endogenously when Cre is not expressed. The design of dual reporters allows us monitor the efficiency and specificity of the recombination. These data are missing in the paper to show how Dat-cre works on the flox-Dnmt3a2/3L, how many unexpected cells are affected by Cre leakage, what’s the percentage of DA neurons targeted.
  2. If there’re endogenously GFP expression, the TH and 5mC immunostaining should be worked with other colors in Fig1 B and C.
  3. The quantification of 5mC normalized by TH staining is not an ideal. More solid way is counting 5mC signal over DNA dye like DAPI here.
  4. Regarding to the slightly difference between genders in cylinder test, whether electrophysiological properties have a similar pattern?
  5. Since striatal DA levels didn’t have significant changes, how about these neurotransmitters level in hypothalamus or piriform cortex found with higher dopaminergic activity?
  6. The word “regulates” in Abstract line 38 and Discussion Line 203 is a bit confusing, I would propose “disturb” or “disrupt”.

Author Response

Dear Reviewer 3,

Many thanks for your comments and suggestions. Please see the attachment as reply.

Best regards,

Authors

Round 2

Reviewer 2 Report

To my original report, the authors have now come up with an explanation that the overexpression system indeed is responsible for hypermethylation, and DNA methylation mediated by Dnmt3a2/3L is able to increase neuronal activity in DA neurons, leading to elevated locomotor activities. The data presented in the rebuttal doesn’t support the claim.

The authors have cited that they have checked the expression of Dnmt3a2 at tissue level by western blot and qRTPCR (Fig.1C, D). The results show an increasing of Dnmt3a2 level in midbrain where the cell bodies of SNc neurons are located.

In figure 1C, the western-blots the wt and Dat/wt loads are not similar. Indeed, for OB, FC and MB lanes, Dat/wt has 25-40% higher beta-actin compared to wt. Any elevated expression of Dnmt3a2 is due to higher cell extract in western-blot.

In figure 1D, the qPCR result shows a modest increase in mRNA for Dnmt3a and Dnmt3L (1.0 to 1.17).

In figure 1E the immune stains (IHC) using anti-5mC is not definitive. Indeed, immunoblots for quantitative increase or decrease of DNA methylation is notoriously inaccurate. A global 5mC determination using LC-MS will be warranted.

Due to the above inaccuracy and lack of additional experiments, I still feel that the author claim is incorrect. It will be highly misleading without additional experiments to claim the role of DNA hypermethylation in the manuscript.

Author Response

Response to Reviewer’s further comments:

We again so appreciate for reviewer’s rigor for mechanism understanding as it implies the current study is so important for the field.  We agree with reviewer’s comment on our current evidence which is not crystal clear regarding the molecular/epigenetic mechanism regarding this significant biological finding. This aligns well with our future research plan, and we take it seriously.

One piece of information we want to share with our reviewer is that we have conducted a systematic in vitro study of Dnmt3a2/Dnmt2L on DNA methylation, gene expression as well as interaction with other key proteins by using the cutting-edge technologies such as whole genome methylation sequencing,  RNA-seq and Co-IP related proteomics (another manuscript).  We are confident, at least at neuronal cell culture,  Dnmt3a2 and Dnmta2/3L do regulate the DNA methylation as well as the neuronal gene expression, and modulate the cell fate.  However, we do understand the difference between in vitro and in vivo

We thus took the advice from reviewer and editor and made a significant change for our conclusion part (last paragraph) as following:

We conclude that DNA methylation mediated by Dnmt3a2/3L is able to increase neuronal activity in DA neurons, leading to elevated locomotor activity. This is a first step in understanding the neuronal effects of epigenetic modification and its behavioral consequences. We are aware the present study has not explicitly elucidated the role of DNA methylation by Dnmt3a2/3L in this significant biological consequence yet. For example, on the tissue level we hardly obtained any significantly synaptic plasticity gene expression change (SFig.4) though we indeed achieved that at the in vitro study, which highly likely due to the low percentage of the dopaminergic neurons among all cell populations in the brain tissues. To further decipher the role of DNA methylation as well as the gene expression regulated by Dnmt3a2/Dnmt3L in neuronal functions and behavioral changes, a catalytic domain mutant (Cys-Ala) control mouse cohort is needed to understand the enzyme activity of Dnmt3a2 (Jia et.al) and the single cell (population) DNA methylation, gene expression and recording are essential to decode the involving epigenetic and molecular mechanism(s). 

Jia D, Jurkowska RZ, Zhang X, Jeltsch A, Cheng X. Structure of Dnmt3a bound to Dnmt3L suggests a model for de novo DNA methylation. Nature. 2007 Sep 13; 449(7159):248-51.

Hope this will eliminate your concern for overstating of our findings at this present study.

Round 3

Reviewer 2 Report

Thank you, that looks good.